# Cryostructuring of Polymeric Systems. 52. Properties, Microstructure and an Example of a Potential Biomedical Use of the Wide-Pore Alginate Cryostructurates

**DOI:** 10.3390/gels5020025

**Published:** 2019-05-09

**Authors:** Natalia D. Zvukova, Tamara P. Klimova, Roman V. Ivanov, Andrei N. Ryabev, Archil V. Tsiskarashvili, Vladimir I. Lozinsky

**Affiliations:** 1A.N. Nesmeyanov Institute of Organoelement Compounds, Russian Academy of Sciences, Vavilov Street, 28, 119991 Moscow, Russia; natalya_zvukova@mail.ru (N.D.Z.); tab@ineos.ac.ru (T.P.K.); erawell@mail.ru (R.V.I.); ryabev@ineos.ac.ru (A.N.R.); 2N.N. Priorov National Medical Research Center of Traumatology and Orthopedics, Ministry of Health of the Russian Federation, Priorov Street, 10., 127299 Moscow, Russia; armed05@mail.ru

**Keywords:** cryostructurates, sodium alginate, calcium alginate, alginic acid, freeze-drying, swelling characteristics, microstructure, antibacterial material

## Abstract

Wide-pore cryostructurates were prepared via freezing sodium alginate aqueous solutions with subsequent ice sublimation from the frozen samples, followed by their incubation in the ethanol solutions of calcium chloride or sulfuric acid, rinsing, and final drying. Such sequence of operations resulted in the calcium alginate or alginic acid sponges, respectively. The swelling degree of the walls of macropores in such matrices decreased with increasing polymer concentration in the initial solution. The dependence of the degree of swelling on the cryogenic processing temperature had a bell-like character with a maximum for the samples formed at −20 °C. According to ^1^H NMR spectroscopy, the content of mobile (non-frozen) water in the frozen water-sodium alginate systems also depended on the initial polymer concentration and freezing temperature. The cryostructurates obtained did not lose their integrity in water, saline, in an acidic medium at pH 2 for at least three weeks. Under alkaline conditions at pH 12 the first signs of dissolution of the Ca-alginate sponge arose only after a week of incubation. Microbiological testing of the model depot form of the antibiotics entrapped in the Ca-alginate cryostructurate demonstrated the efficiency of this system as the antibacterial material.

## 1. Introduction

Polymeric cryostructurates are the macroporous materials of spongy morphology, which are formed as a result of freezing the solutions of macromolecular precursors or certain gels with subsequent removal of the crystallized solvent phase without thawing the system as a whole [1]. In this context, two types of cryostructurates can be distinguished [2]:

(**I**) In the first case, a cryostructurate is produced by freezing a molecular or colloidal polymer solution followed by removing the polycrystals of the solvent by sublimation or cryoextraction. As a rule, the resulting porous material is soluble. Therefore, to convert it into an insoluble form, either additional chains crosslinking (chemical or radiation) or the respective chemical modification of functional groups of the polymer is required. 

(**II**) Upon the preparation of the second type of cryostructurates, the already-formed gel is frozen and then the solvent crystals are removed, most often via sublimation, as, for example, in the production of so-called “freeze-dried food” or some lyophilized pharmaceuticals [3,4]. 

In both cases, no gel formation occurs in the frozen samples, which distinguishes cryostructurates from cryogels, the formation of a three-dimensional spatial network of which, on the contrary, is triggered by freezing the precursor solution and proceeds exactly in a frozen medium [1,2,5,6].

Polymeric cryostructurates are of great interest both as objects of basic research and as materials for applications. The physico-chemical properties and macroporous morphology of various cryostructurates are determined by many factors, including the nature of the precursors, their concentration in the feed system (before cryogenic processing), the conditions of freezing, and removal of the crystalline phase, the methods of imparting insolubility to cryostructurates of the above-indicated type (**I**) [1,2,3,4,7,8,9,10,11]. 

For instance, these polymeric matrices include polyelectrolyte cryostructurates formed according to a three-stage scheme: freezing a solution of polyelectrolyte—removing the crystallized solvent phase—transforming the polymer into an insoluble form (either using the crosslinking counterions or by recharging the ionogenic groups of macromolecular chains) [5,12,13]. In this context, cryostructurates and cryogels based on natural polymers such as ionic polysaccharides, in particular, alginic acid (the copolymer of β-d-mannuronic and α-l-guluronic acids) and its salts, attract the increasing attention during recent years, especially upon the elaboration of biomedical and biotechnological materials [14,15,16,17,18,19,20,21,22], in food technology [23], environment protection [24], etc. This is determined, firstly, by the availability and low cost of alginates; secondly, by their biocompatibility, nontoxicity, and non-immunogenicity; thirdly, by the good gel-forming properties, which, like the characteristics of the obtained gel materials, can be controlled by choosing a polymer with the required molecular weight, ratio, and blockiness of monomeric units, as well as the nature and concentration of the crosslinking counterions [25,26,27,28].

Conventional alginate hydrogels are formed upon the introduction of divalent or trivalent metal cations into the aqueous solutions of the sodium salt of this polyelectrolyte [25,26,27,28,29,30,31]. In turn, alginic acid gels are formed when the alginate solution is acidified below the pK_a_ value of the uronic units of this biopolymer [32]. Since the ionotropic gelation of alginates is a very fast process that proceeds almost immediately after the components are mixed, it is technically difficult to carry out the gelation exactly in a frozen system (this is precisely what is needed to obtain cryogels [5]), because the formation of a spatial polymer network occurs earlier than the system freezes. Simultaneously, the forming solvent crystals are capable of disrupting the integrity of the low-strength network. Therefore, alginate cryogels can be obtained from solutions only with a rather low concentration of precursors. 

The alternatives are the type (**I**) alginate cryostructurates, when the content of components in the system depends only on the solubility of the alginic acid salt used to prepare the starting solution of this polyelectrolyte. It was shown that this approach is very effective for obtaining spongy alginate cryostructurates [12] that were used as a polymeric base of the wide-pore scaffolds for the three-dimensional cultivation, differentiation, and cryopreservation of stem cells [33,34], as well as were employed as the carriers for the medicamental nano-forms [35]. The latter studies revealed a rather prospective applied potential of such alginate sponges for the elaboration of wound dressings (these experiments are in progress now; their results will be reported in the subsequent publication). Therefore, there was the necessity for a systematic study of the factors influencing the properties and macroporous morphology of such alginate cryostructurates with the aim of fine controlling their characteristics. This opens the routes for the experimentally based selection of these materials in solving specific problems of their use, e.g., as the polymer carriers for drugs, in particular, antibiotics. Exactly such research was the goal of this work, and we mainly interested in the properties of wet and swollen alginate sponges since upon their use as the wound dressings, where the materials operate in aqueous media.

## 2. Results and Discussion

### 2.1. Influence of Formation Conditions on the Properties of Ca-Alginate and Alginic Acid Cryostructurates

As it is known, the factors mainly influencing the physicochemical properties of any polymeric cryostructurates of the type (**I**) essentially include the concentration of the macromolecular precursor in the feed solution and the temperature conditions of its cryogenic treatment [1,2,5,8,9,10]. For this reason, we studied the dependence of the characteristics of alginate and alginic cryostructurates on these parameters. The schematic diagram of their formation is illustrated in Figure 1, and the method for their preparation is described in the “Experimental”.

Preliminary experiments showed that calcium-alginate cryostructurates (Ca-ACRS) and alginic acid cryostructurates (H-ACRS) that possessed the properties convenient for further research can be formed from the sodium alginate (Na-ALG) aqueous solutions in the range of polymer concentrations from 10 to 50 g/L. At a lower initial polysaccharide concentration, rather weak cryostructurates were formed that had low strength in the swollen state. On the other hand, when the concentration of Na-ALG was above the indicated range, a considerable increase in the viscosity of the stock polymer solution took place thus significantly complicating its dosing. As for the freezing conditions of the Na-ALG solutions, at negative temperatures above −10 °C the system often did not crystallize because of the supercooling effects. In turn, the samples freezing below −30 °C caused a marked decrease in the pore cross-section of the resultant cryostructurates, especially prepared from the solutions with a polymer concentration of 40–50 g/L. Therefore, in the course of further research, the feed solutions of sodium alginate were frozen at a temperature over the range from −10 to −30 °C.

After ice sublimation from the frozen samples, the resultant sodium alginate cryostructurate (Na-ACRS) was converted either to calcium salt (Ca-ACRS) or to acidic form (H-ACRS) to impart them the insolubility in water. This treatment was carried out in an ethanol medium which wetted the polymer well, but did not dissolve it, i.e., the replacement of Na^+^ counterions for Ca^2+^ or H^+^ cations, respectively, actually occurred in the solid phase without disturbing the spongy morphology of the material. In contrast, if to carry out such ion-exchange processes in an aqueous medium, an undesirable swelling or even partial dissolution of the polymer can occur. In turn, in ethanol, thus “insolubilized” Ca-ACRS and H-ACRS cryostructurates were tough, while after replacing ethanol with water and achieving equilibrium swelling, they became rather soft sponges. As a result, a free fluid from the interconnected wide pores could be pressed out, for example, on a glass filter under vacuum (see “Experimental”). This property of the water-swollen Ca-alginate and alginic acid cryostructurates, i.e., squeezing of capillary liquid by mechanical action, does not allow the correct measurement of the physico-mechanical characteristics of the swollen matrices, but it makes possible to evaluate the degree of swelling (*S*_w/w_, g H_2_O/g polymer) of their polymeric phase, i.e., the walls of macropores. This procedure was successfully employed earlier for various cryogenically structured macroporous spongy materials based on polymers of both synthetic [5,36,37,38,39,40,41], and natural (chitosan [42,43,44], agarose [45,46], albumin [9,47], gelatin [10,48], etc.) origin. In the case of alginate-containing cryostructurates, the *S*_w/w_ values were determined for samples formed from the solutions of different Na-ALG concentrations. The data obtained in these experiments are shown in the graphs of Figure 2 in the form of the dependences of the swelling degree values for Ca-ACRS (a) and H-ACRS (b) on the polymer concentration in the solution before its freezing at −20 °C.

The common trend for both Ca-ACRS and N-ACRS was a decrease in the swelling degree of the polymeric phase in these spongy cryostructurates as the initial concentration of Na-ALG increased. This indicates that the use of a more concentrated solution of a polysaccharide precursor results in a heterophase material with denser walls of macropores, where the polymer chains are cross-linked with calcium ions in Ca-ACRS or via the hydrogen bonds such as “carboxyl-hydroxyl” and “hydroxyl-hydroxyl” in H-ACRS. In the latter case, alginic acid sponges, prepared according to the scheme of Figure 1 from a 10 g/L Na-ALG solution, were rather weak. Therefore, it was not possible to determine correctly their swelling degree. In turn, the absolute values of the parameter *S*_w/w_ equal to 5.7–3.0 g H_2_O per 1 g of dry matter in the polymeric phase of Ca-alginate sponges formed from the 20–50 g/L Na-ALG solutions (Figure 2a) were approximately 20–30% higher (2.5–4.1 g/g) than the values for the corresponding H-ACRS sponges (Figure 2b). This fact indicates a slightly greater hydration of the walls of macropores in the Ca-ACRS, i.e., the affinity of water molecules to calcium alginate is, obviously, higher than that in respect with the protonated form of alginic acid. 

It is well-recognized that one of the key factors determining the properties and porous morphology of polymeric matrices formed using the cryogenic structuring approaches is the temperature conditions of freezing of the initial liquid system [1,2,3,4,5,6,49,50,51,52]. Therefore, we evaluated the influence of freezing temperature of the sodium alginate aqueous solution on the degree of swelling of the polymeric phase in the resultant cryostructurates. The data obtained for the Ca-ACRS samples formed from a polysaccharide solution with [Na-ALG] = 30 g/L are shown in Figure 3.

It turned out that such temperature dependence had a bell-shaped character, when the walls of macropores of a spongy cryostructurate formed by freezing at −20 °C swelled to the greatest extent (*S*_w/w_ ≈ 4.9 g/g). The polymeric phase of the wide-pore samples prepared via freezing at both above (i.e., −15 and −10 °C) and below (−25 and −30 °C) this temperature swelled to a lesser extent (3.6–3.8 g/g). The similar extreme nature of the temperature effect of the frozen system on the osmotic properties (swelling) of the macropore walls in the cryogenically structured heterophase matrices was previously described, for example, for chitosan cryogels [53], which evidence the very close temperature effects. The reason is the competition of oppositely-directed processes [1,5]. In particular, the lower is the temperature of the frozen solvent-polymer system, the higher is the concentration of the polymer in the volume of a so-called unfrozen liquid microphase (UFLMP) [54,55] due to the freezing out a larger solvent volume. Therefore, there is a higher probability of intermolecular contacts, as well as overlapping of the chains with the formation of topological entanglements, which contributes to the compaction of the polymer phase. As a result, the walls of macropores in the already-formed cryostructurate will swell weaker at positive temperatures. On the other hand, the lower is the temperature of the frozen sample, the higher is the viscosity of its UFLMP, which significantly limits the mobility of macromolecules and their segments, thus preventing the compaction of the polymeric phase in the “future” spongy cryostructurate. This kind of competition results in bell-shaped dependencies of the properties of cryogenically structured polymer matrices on the temperature of cryogenic treatment of the initial systems in general and the dependence of the degree of swelling of Ca-ACRS samples on the freezing temperature of the Na-ALG solution in particular (Figure 3). In turn, regarding the potential biomedical application of the drug-loaded alginate sponges, the ratio of the loaded drug amount inside the capillary pores and within the polymeric gel phase of pore walls will depend on their (the walls) swelling extent. This ratio should influence on the drug release kinetics—quick from the intrapore space and somewhat slower from the pore walls.

In this regard, interesting data were obtained in the NMR measurements of the mobile water content in the non-deeply frozen (−10, −20 and −30 °C) sodium alginate solutions of different initial polymer concentration (20, 30, and 35 g/L). The results of these experiments are given in Figure 4 as the temperature dependence of the parameter α—the ratio of the areas of the signals for the water protons in the frozen samples and the signals recorded at room temperature for the solutions of the same polymer concentrations (see “Experimental”).

The content of nonfrozen water at different subzero temperatures for the initially equiconcentrated Na-ALG systems systematically decreased with lowering the temperature from −10 to −30°C, which is understandable, since the lower is the temperature of the frozen sample, the greater the amount of solvent crystallizes. However, in relation to the influence of polymer concentration in the frozen systems on the ***α*** parameter no direct dependence was observed. At three negative temperatures used the lowest content of nonfrozen water, which retains mobility according to the NMR data, was found in the case of a system with an initial polymer concentration of 30 g/L (***2***; Figure 4). In the samples with an initial Na-ALG concentration of 20 and 35 g/L (***1*** and ***3***, respectively; Figure 4), at each minus temperature the content of the nonfrozen water was higher.

There is still not enough data to interpret this fact accurately, but at a qualitative level, such concentration dependence (with a minimum) indicates, most probably, the competitive nature of the processes that ultimately determine the phase equilibrium in frozen aqueous Na-ALG solutions. In similar moderately frozen systems, one of the key factors is the viscosity of the UFLMP, since the higher is the initial polymer concentration, the higher will be the viscosity of the unfrozen fraction in a macroscopically frozen sample because the dissolved substances are concentrated in the UFLMP. It is known that the crystallization of the solvent during its freezing in highly viscous media occurs with significant hindrances [56]. On the other hand, it cannot be ruled out that the probability of the glass transition for some part of the UFLMP increases with lowering the temperature, thereby reducing the content of mobile water, and this factor is also included in the “competition” with other factors. In any case, the experimental data testify to the “consequences” of such competition, which is reflected in the properties of cryostructurates formed on the basis of solutions of different concentrations of the polymer precursor and at different temperatures of the cryogenic stage.

### 2.2. Macroporous Morphology of Alginate Cryostructurates

As it is well-known [1,2,3,4,5,6,36,37,39,42,45,46,47,49,50,51,57,58], the temperature conditions of the cryostructuring of polymer systems exerts a significant influence on the texture of the resulting heterophase matrices, as well as on the volume fraction, size, and shape of their macropores. Therefore, we studied the features of the macroporous morphology of alginate cryostructurates formed at the same freezing temperatures of the polysaccharide solutions in comparison with the samples the osmotic characteristics of which (Figure 3) were discussed above. As an example, Figure 5 shows the micrographs (optical microscope) of Ca-ACRSs prepared by freezing the 30 g/L Na-ALG aqueous solution at −10, −20, and −30 °C.

An analysis of such images indicates that the size and shape of macropores in these alginate matrices are not entirely dependent directly on the freezing temperature. Thus, the pores in the Ca-ACRS sponge formed at −10 °C had a cross-section from 65 to 160 μm (Figure 5a), and in a cryostructurate formed at −20 °C, this interval was from 40 to 110 μm (Figure 5b), i.e., with lowering the freezing temperature a well-known regularity was observed, which is stipulated by a diminution of the porogen (ice polycrystals) size with a deeper cooling of the system under freezing [16,57,58]. At the same time, at even lower freezing temperature, in particular, at −30 °C, the cryogenically-structured samples with very irregular porosity with respect to the geometry of the macropores and their sizes were formed (Figure 5c). This result indicates a sophisticated nature of the “low-temperature” freezing even of a rather thin layer (~2 mm) of sodium alginate solution being in a plastic Petri dish placed on a cooled plate (see “Experimental”). In this context, it could not be excluded that along with the ice crystallization within the freezing sample, a partial glass transition processes can (as it was pointed out above) occur upon rapid cooling [58], thus interfering the movement of the solvent crystallization fronts. It is clear that this assumption requires additional study, which was not the task of the present work.

The samples for the microscopy observation were prepared in the shape of flat disks, and the micrographs in the Figure 5a–c display the texture of the bottom surface of each disk, which, when the stock solution of polysaccharide was frozen, contacted with the bottom of the cooled Petri dish. Therefore, the temperature gradient had a vertical direction, when the solvent crystallization front moved upwards. At the same time, due to the “branched” nature of the growth of ice crystals [56,59,60], the pores in the upper part of the flat cryostructurates, firstly, greatly change their shape and become significantly larger than those in the lower part, and, secondly, the overall texture becomes less regular. Exactly the same (at the qualitative level) pattern of certain differences in the macroporous morphology of the upper and lower regions of flat freeze-structured polymer matrices was observed for cryogels and cryostructurates based on agarose [45,61], serum albumin [9,47], gelatin [10], poly(vinyl alcohol) [62], and some others [49,50,51,63,64,65]. In the case of alginate cryostructurates, a very similar tendency was also met. For example, it is well-detected when comparing the micrographs in Figure 5b,d, showing the morphology of the lower and upper surfaces, respectively, of the Ca-ACRS disc formed by freezing at −20 °C.

### 2.3. Behavior of Alginate Cryostructurates in Aqueous Media of Different Composition

As it was already noted, alginate-containing cryostructurates are promising materials in terms of their use for solving a number of biomedical and biotechnological problems. In such cases, the “operation” of these matrices usually takes place in aqueous media of various composition. Therefore, it is important to have the data on the stability or, conversely, the destruction of such polysaccharide sponges in the respective liquids. To this end, we observed the preservation of the integrity of the Ca-ACRS disks when they were incubated for three weeks in one of the following fluids:In the deionized water (control);In the 0.15 M aqueous solution of NaCl (saline), i.e., in the conditions promoting the shift of the ion exchange equilibrium towards the replacement of calcium ions in Ca-alginate by Na^+^-ions due to the excess of the latter ones and, as a consequence, the possible solubilization of the polymer;In the 0.01 N HCl solution at pH 2, where the transformation of Ca-ACRS to H-ACRS should take place, but without dissolving the polymer framework of the cryostructurate;In the 0.01 N NaOH solution at pH 12, i.e., under the conditions where conventional Ca-alginate hydrogels are dissolved rather quickly [16].

The results obtained in these experiments are shown in the photographs of Figure 6, where one can see the appearance of the test samples during their incubation in the above-indicated milieus. As a result, it was found that these spongy cryostructurates are stable not only in water (Figure 6a–d) and acidic medium (Figure 6i–l), where neither Ca-alginate nor alginic acid dissolve, but also in an excess of Na^+^-ions in physiological solution (Figure 6e–h). Only in an alkaline medium at pH 12 a very slow (starting approximately from the middle of the second week of the experiment) solubilization of the polymer material was observed (Figure 6m–p).

Such a high stability of the Ca-ACRS sponges, as well as the H-ACRS matrices in an acidic medium, formed according to the sequence of operations we used (Figure 1), most likely is due to the fact that crosslinking of the alginate chains with calcium ions or intermolecular hydrogen bonding of the uronic acid groups occurred within the bulk of the highly concentrated polymeric walls of macropores of the spongy matrix poorly swollen (wetted) in ethanol. In this case in the condensed phase of the walls of macropores, the functional groups of the macromolecular chains are very close together, which should contribute to the formation of a dense frequently crosslinked polymeric network. In contrast, in the usual cases of ionotropic alginate gelation, when the crosslinking process proceeds in solution [14,15,16,25,26,27,28] where the solvated polymer chains are distant from each other, only some fraction of their functional groups can be involved in the intermolecular binding.

It may also be noted that at the qualitative level there is a similar tendency, i.e., the maximum possible degree of crosslinking of macromolecules in the condensed phase of the walls of macropores, was found in the case of the cryogenically structured protein sponges formed by freezing aqueous solutions of serum albumin, their subsequent freeze-drying and further chemical tanning with the carbodiimide reagent in an ethanol medium [9]. Hence, one can draw the conclusion that the preparation of the wide-pore polymer matrices according to the scheme for the formation of the type (**I**) cryostructurates (Figure 1) makes it possible to increase the stability of the polymer phase of such spongy materials with respect to various solubilizing factors.

### 2.4. In Vitro Testing of Ca-Alginate Cryostructures as a Polymeric Carrier for the “Depot Form” of Antibiotics

In modern medical practice, in particular, in the treatment of wounds and burns, spongy coatings containing medicinal agents, frequently called “depot forms” of appropriate medicines, are widely employed [66,67]. Such wide-pore materials perform several functions: Absorption of liquid exuded by a wound, protection of the wound from contact with the environment, as well as the delivery of the medication to the lesion. Since certain alginate materials are permitted for medical use [16,17,68,69,70,71], we conducted a primary in vitro assessment of the possible biomedical potential of the Ca-ACRS/vancomycin depot formulations using the standard diffusion test (see “Experimental”). To assess the antimicrobial activity of such formulation, a culture of methicillin-resistant microorganism belonging to the species of *Staphylococcus aureus* was used. Figure 7 shows an example of such testing.

The technique used in these experiments makes it possible to visualize the release of a drug (in our case, an antibiotic) from the polymeric carrier [35]. As a result, a growth suppression zone of bacterial culture colonies on an agar nutrient medium is formed. Such zone of clearing is visible well in the photograph of Figure 7 around the sector of the Ca-alginate sponge loaded with vancomycin (~93 mg per a spongy disk of 38 mm in diameter and ~2 mm thickness). The sector was cut from the disc. In turn, on the control Petri dish with the same staphylococcal colonies, in the center of which was located a fragment of a spongy Ca-ACSR without an antibiotic, the formation of the growth suppression zone was not observed at all. Thus, in these experiments, the performance of the alginate sponges as the polymeric carriers for antibiotics was shown, which indicates the prospects for the biomedical use of similar depot forms of antibacterial medications.

## 3. Conclusions

Currently, alginic acid and its salts are widely employed in various fields of human activity, especially in food and biomedical technologies. In the former case, these ionic polysaccharides are used as the functional ingredients and the polymer base of some food forms (jelly products), in the latter case–as the biomedical materials (covers on wounds and burns, scaffolds for cell culture, etc.). In this regard, of a certain scientific and applied interest is the preparation of alginate materials using the methods of cryostructuring of macromolecular systems. This is because such approach allows obtaining the wide-pore polymer matrices with a developed system of interconnected large pores. In the present work, this method was used to create spongy alginic acid and calcium-alginate cryostructurates that were prepared by freezing at −10…−30 °C sodium alginate aqueous solutions (10–50 g/L), followed by sublimation of ice from the frozen samples, the treatment of freeze-dried spongy matrices with, respectively, ethanol solution of sulfuric acid or calcium chloride, washing the resultant sponges and their final drying in vacuum. It was found that the degree of swelling of the walls of macropores in such matrices decreases with increasing the polymer concentration in the initial solution. With that, the dependence of the degree of swelling on the cryogenic treatment temperature has a bell-like character with a maximum for the samples formed by freezing at −20 °C. ^1^H NMR spectroscopic studies of frozen sodium alginate solutions showed that the content of mobile (non-frozen) water therein also depends on the initial concentration of the polymer and the temperature of the cryogenic treatment. The cryostructurates obtained in this work were stable in aqueous media, i.e., for at least three weeks the sponges did not lose their integrity in water, saline, and in an acidic medium at pH 2. In an alkaline medium at pH 12, the first signs of dissolution of the Ca-alginate sponge arose only after a week-long incubation. Microbiological testing of a model depot form of an antibiotic (vancomycin) entrapped in the wide porous Ca-alginate cryostructurate demonstrated the efficiency of this composite system thus testifying the prospects of creating such depot forms of various medicinal agents using cryogenically structured polymer carrier based on this biocompatible and affordable polysaccharide.

## 4. Experimental

### 4.1. Materials

The following substances and preparations were used without additional purification: Sodium alginate (Na-ALG) (BDH Chemicals Ltd., Poole, UK) (MM 150 kDa; the viscosity of the 1% aqueous solution 0.63 Pa·s (23 °C); the content of mannuronic blocks was 30%, the content of guluronic blocks was 20%, and the content of the mixed sequence blocks was 50% [26]), calcium chloride (anhydrous, Panreac, Barcelona, Spain), vancomycin (Teva Pharmaceutical Industries Ltd., Petah Tikva, Israel), sulfuric acid (chemically pure, IREA-2000 Ltd., Moscow, Russian), 95% ethyl alcohol (Khimmed, Moscow, Russian), sodium hydroxide (Chemapol, Prague, Czech Republic), hydrochloric acid 38% (Reakhim, Moscow, Russian), and physiological saline (JSC Biochemist, Moscow, Russian). Deionized water was used to prepare polymer solutions. 

### 4.2. Preparation of Alginate and Alginic Cryostructurates

The cryostructurates were formed from the aqueous solutions of Na-ALG (10–50 g/L) that were initially filtered under vacuum through a porous glass filter, and then treated for 30 min in an ultrasonic bath UM-1 (Unitra-Unima, Warsaw, Poland) for removing air bubbles. Next, the polymer solution in portions of 2.25 g was poured into plastic Petri dishes (Medpolymer, Moscow, Russian) with an inner diameter of 38 mm. The dishes were placed in the chamber of a Proline RP 1840 ultracryostat (Lauda, Königshofen, Germany) and kept for 1 h to freeze the liquid at a pre-set temperature (−10; −15; −20; −25; or −30 °С), after which the ice was sublimated in vacuo (0.040 mbar) for 18 h using an Alpha 1-2 LD plus freeze-drier (Martin Christ, Osterode am Harz, Germany). Then, in order to obtain the water-insoluble Ca-alginate cryostructurates (Ca-ACRS in Figure 1), the freeze-dried spongy Na-ALG discs were immersed for 24 h in a saturated ethanol solution of CaCl_2_, further rinsed with an excess of alcohol and finally dried in vacuo at 1 mm Hg. For obtaining of the water-insoluble alginic acid cryostructurates (H-ACRS in Figure 1) the spongy Na-ALG discs were immersed for 24 h in a 0.1 M ethanol solution of H_2_SO_4_, rinsed with alcohol, and dried in vacuo like the Ca-ACRS samples.

### 4.3. Characterization of the Cryostructurates

The swelling degree of the polymeric phase (the walls of macropores) in the wide-pore Ca-alginate and alginic acid cryostructurates was determined by the gravimetric method. To do this, free liquid was removed from the water-swollen sponges on a glass filter under vacuum of a water-jet pump. Thus obtained “pressed-out” wet samples were weighed and then dried at 110 °C to a constant weight in an air thermostat SNOL 24/200 (AB Utenos Elektrotechnika, Utena, Lithuania). The degree of swelling by weight (*S*_w/w_) was calculated by the formula:*S*_w/w_ = (*m*_wet_ − *m*_dry_)/*m*_dry_ (g H_2_O/g polymer)(1)
where *m*_wet_ is the mass of the “pressed-out” wet sample, *m*_dry_ is the mass of the dry sample.

When studying the stability of alginate cryostructurates in different liquid media, each of the Ca-ACRS samples was kept in one of the following solutions: 0.15 M NaCl, 0.01 N NaOH, 0.01 N HCl, and H_2_O. The liquid phase was replaced with a fresh portion weekly. The state of cryostructurates was recorded using an α-750 digital photocamera (Sony, Tokyo, Japan).

The macroporous morphology of alginate cryostructures was studied using an Eclipse 55i optical microscope (Nikon, Tokyo, Japan) equipped with a digital image recording system.

### 4.4. NMR Studies

The quantitative measurements of the unfrozen water fraction in frozen Na-ALG solutions was performed by the ^1^H NMR method using an AMX TM 300 spectrometer (Brüker, Hardt, Germany) at an operating frequency of 313 MHz. Aqueous solution of the polymer (20, 30, or 35 g/L) was placed in a polyethylene capillary (internal diameter 0.85 mm) with a sealed lower end. The capillary was put in a standard NMR ampoule containing deutero-acetone. The signals of protons of the solvent (water) were recorded at −10, −20, and −30 °C and their areas were found. Then, their ratio (α) to the areas of the proton signals of Na-ALG solutions of the same concentrations recorded at room temperature was calculated and expressed as a percentage.

### 4.5. Microbiological Tests

To load the polysaccharide cryostucturates with the antibiotic, dry Ca-ACRS sponges were placed in an aqueous solution of vancomycin (50 g/L) and incubated for 24 h at 4 °C with periodical stirring. Then the swollen sponges were frozen at −20 °C and freeze-dried. The amount of the antibiotic in thus prepared sponge has been determined gravimetrically as the difference in the weight of dry sponge before its loading with vancomycin and the weight after loading and freeze-drying. This amount turned out to be 93.4 ± 0.8 mg per a spongy disk of 38 mm in diameter and ~2 mm thickness. The antimicrobial activity of the already obtained antibiotic “depot forms” obtained was determined by the diffusion method according to the protocol described elsewhere [35] using the *Staphylococcus aureus* DMBV test culture from the collection of microorganisms of the National Medical Research Center of Traumatology and Orthopedics.

## Figures and Tables

**Figure 1 gels-05-00025-f001:**
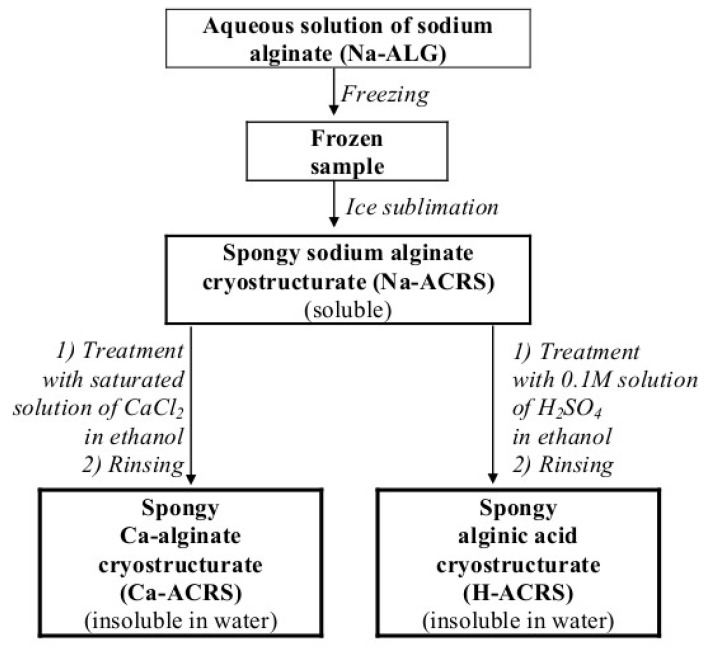
Schematic diagram of the preparation of Ca-alginate and alginic acid cryostructurates.

**Figure 2 gels-05-00025-f002:**
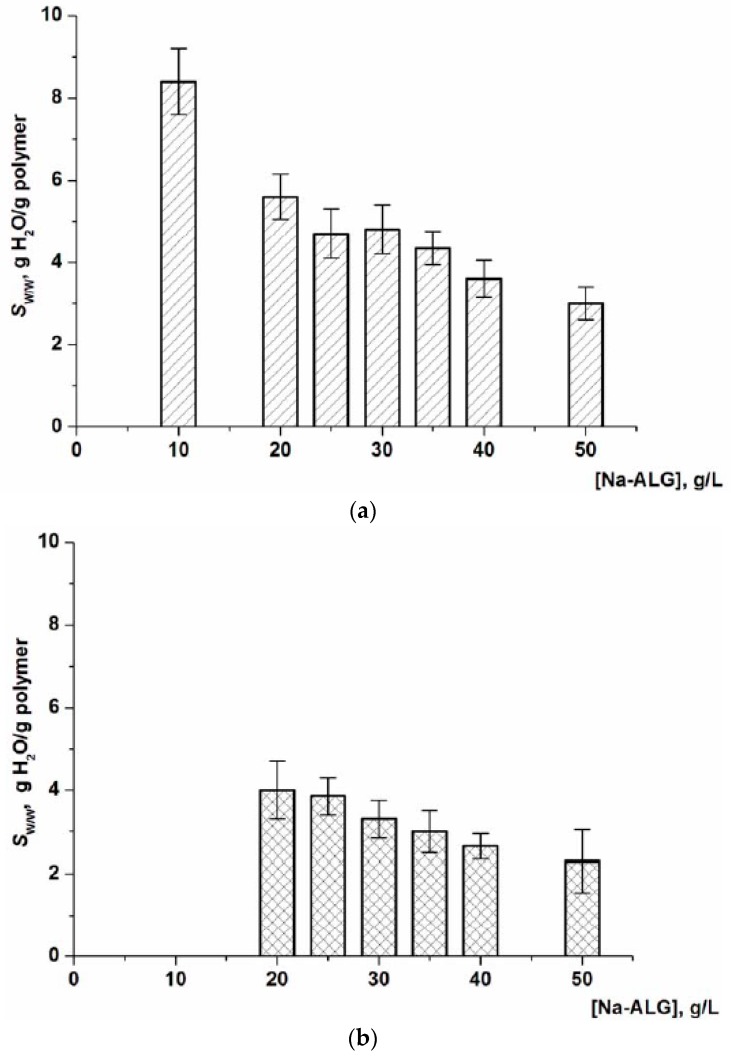
The swelling degree of the polymeric matter in the walls of macropores in Ca-ACRS (**a**) and H-ACRS (**b**) samples formed by freezing at –20 °C of aqueous polymer solutions of different Na-ALG concentrations.

**Figure 3 gels-05-00025-f003:**
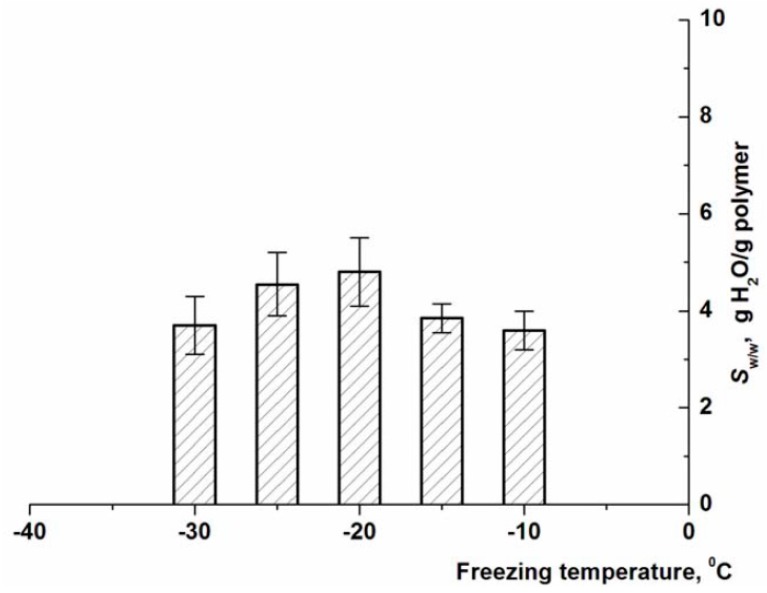
The swelling degree of the polymeric matter in the walls of macropores of Ca-ACRS samples formed by freezing the Na-ALG solution (30 g/L) at different negative temperatures.

**Figure 4 gels-05-00025-f004:**
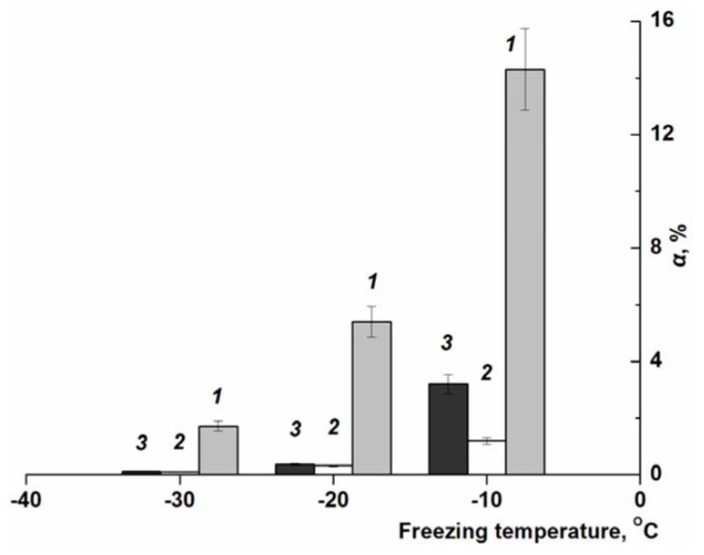
Relative content (α) of mobile (nonfrozen) water in the macroscopically frozen aqueous Na-ALG solutions of different concentrations of the polymer: 20 (***1***), 30 (***2***), and 35 (***3***) g/L, when the samples have been frozen at various minus temperatures.

**Figure 5 gels-05-00025-f005:**
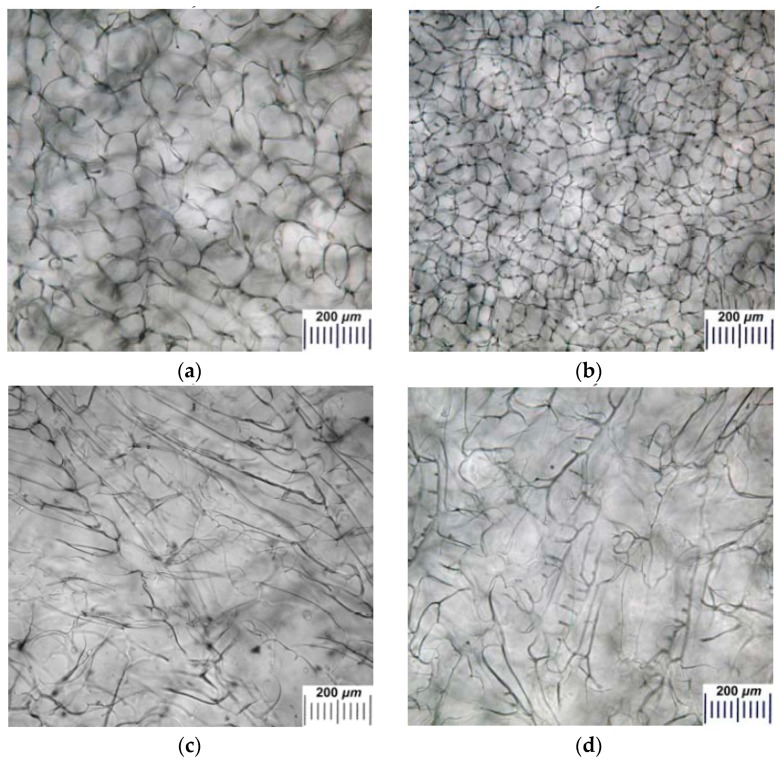
Micrographs (optical microscope) of the water-swollen Ca-ACRS samples formed on the basis of an aqueous Na-ALG solution (30 g/L) by its freezing at the following temperatures: −10 °C (**a**) the bottom surface of the disk; −20 °C (**b**) the bottom surface of the disk and (**d**) the top surface disk; and −30 °C (**c**) the bottom surface of the disk.

**Figure 6 gels-05-00025-f006:**
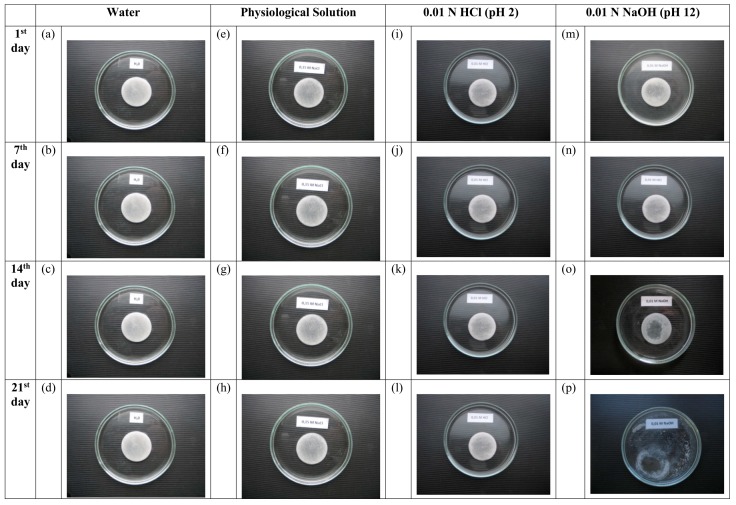
Appearance of the Ca-ACRS disks prepared by freezing at −20 °C of an aqueous Na-ALG solution (30 g/L), when the resultant disks were incubated in the liquid media of different composition. The symbols (**a**–**p**) indicate the particular samples are under discussion within the text.

**Figure 7 gels-05-00025-f007:**
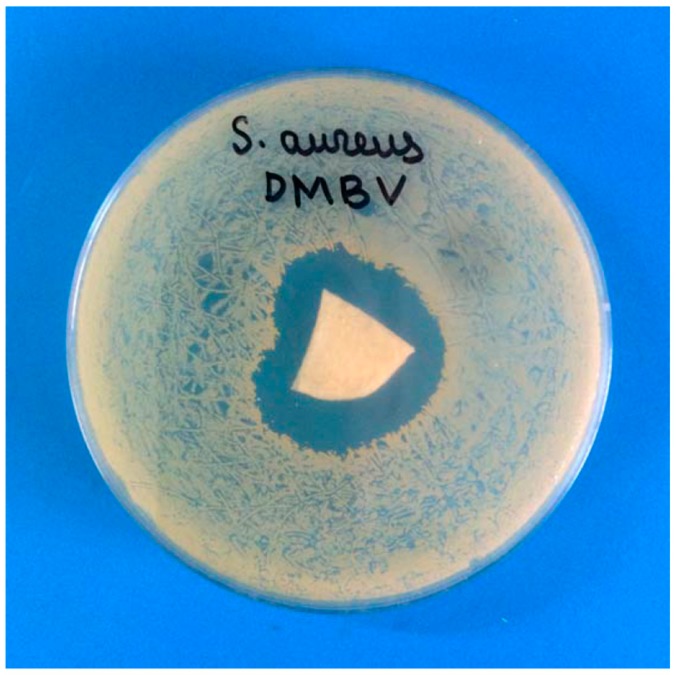
The result of the diffusion test, which demonstrates the formation of the zone of inhibition of the growth of bacteria *Staphylococcus aureus* caused by the release of vancomycin from the Ca-alginate sponge.

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
