# Peer review of "Cryostructuring of Polymeric Systems. 52. Properties, Microstructure and an Example of a Potential Biomedical Use of the Wide-Pore Alginate Cryostructurates"

_gels, 2019, doi:10.3390/gels5020025_

Round 1

Reviewer 1 Report

In this manuscript, the authors reported a study title with ‘Cryostructuring of polymeric systems. 52. Properties, microstructure and an example of a potential biomedical use of the wide-pore alginate cryostructurates’ on Gels.

They prepared wide-pore cryostructurates via freezing sodium alginate aqueous solutions with subsequent ice sublimation from the frozen samples, followed by their incubation in the ethanol solutions of calcium chloride or sulfuric acid, rinsing and final drying. They performed several characterization experiments and showed that the efficiency of this system can be used as an anti-bacterial material.

Below is a list of comments/suggestions to improve the paper.

1.     Please add more details about the aim of this study in the Introduction part.

2.     Almost half of the references were published before 2010. Please add more recent papers.

3.     You can modify x axis of Figure 2, Figure 3 to clarify of graphs.

4.     Why did you prefer to cut the disk?

5. Please perform swelling behaviors of disks at different media (water, physcological, acidic and basic solutions) in an hour (such as 30 sec, 1 min, 5 min, 15 min, 30 min, 45 min, 1 h) and add a graph for comparison.

Author Response

Below is a list of comments/suggestions to improve the paper.

1.     Please add more details about the aim of this study in the Introduction part.

We added the requested information in the ‘Introduction’ (indicated by red color)

2.     Almost half of the references were published before 2010. Please add more recent papers.

In accordance to the recommendation of the reviewer we added 6 new references published in 2016-2019 yrs. At the same time, we think that the suitability of the respective work for its quotation mainly depends on the scientific value of such particular article rather than on the year of its publication.

3.     You can modify x axis of Figure 2, Figure 3 to clarify of graphs.

“X” axes in the graphs of Figure 2 are the values of sodium alginate (Na-ALG) concentration in the initial solution of this polysaccharide prior to its freezing and freeze-drying. Square brackets are the standard chemical symbol meaning ‘concentration’. As for the “x” axis of Figure 3, one can see very simple words: ‘Freezing temperature, oC’ and nothing else. The values of negative temperatures are given from the right to the left, i.e. toward their decreasing. So, it is unclear how it will be possible to additionally clarify this axis.

4.     Why did you prefer to cut the disk?

This was done because the whole disk (about 4  cm in dia.) was too large for the microbiological test carried out in the Petri dishes that were in our disposal, since relatively large free surface with grown bacteria are usually required. The result obtained with a sector of the disk (Fig. 7) turned out to be evident.

5. Please perform swelling behaviors of disks at different media (water, physcological, acidic and basic solutions) in an hour (such as 30 sec, 1 min, 5 min, 15 min, 30 min, 45 min, 1 h) and add a graph for comparison.

The following phrase has been added to the revised manuscript at the end of ‘Introduction’ section: “… we mainly interested in the properties of wet and swollen alginate sponges since upon their use as the wound dressings, the materials operate in aqueous media”. Therefore, the studies of the early swelling stages for the alginate sponges were out of the tasks within the framework of this particular work. However, the present study is not the finish of our activity in this field, so, when it will be necessary, we will carry out the experiments proposed by the reviewer (many thanks for this suggestion). 

Reviewer 2 Report

Review for manuscript ID gels-484548

The article entitled "Cryostructuring of polymeric systems. 52. Properties, microstructure and an example of a potential biomedical use of the wide-pore alginate cryostructurates " described the synthesis, analysis and uses of alginate cryostructurates thus resulting in calcium alginate or alginic acid sponges.

This work, while presenting very interesting aspects, as well as nice future developments, needs to be greatly improved before being published in gels, as the characterization is often more qualitative than quantitative, and some complementary data is needed. Therefore, I consider that this article could be accepted for publication in Gels after a major revision is carried out.

Here are several remarks to be considered:

Results and discussion

- p. 3, l. 10-14: "On the other hand, when the concentration of Na-ALG was above the specified range, a considerable increase in the viscosity of the stock polymer solution took place thus significantly complicating its dosing": What is this specified range? Besides, in numerous publications, when synthesizing alginate gels, foams or macroporous structures,... the starting aqueous alginate solutions concentrations are more often ranging between 1-5wt%, whereas you chose to use 0.1-0.5wt%, which is very low, and probably explains the very weak cryostructurates obtained at 0.1%. What is significantly more "complicated" to dose when using higher concentrations?

- p.3, l.27-28: " In turn, in ethanol, thus “tanned” Ca-ACRS and H-ACRS cryostructurates were tough, ": What does a "tanned" Ca-ARS mean exactly?

- p.3, l. 29: " As a result, a free fluid from the interconnected wide pores could be pressed out, for example, on a glass filter under vacuum (see ‘Experimental’)." : I do not understand, after reading the experimental section, how you can write that you have dried materials that still contain liquid..."This property of the water-swollen Ca-alginate and alginic acid cryostructurates, i.e. squeezing of capillary liquid by mechanical action, does not allow the correct measurement of the physico-mechanical characteristics of the swollen matrices, but it makes possible to evaluate the degree of swelling (Sw/w, g H2O/g polymer) of their polymeric phase, i.e. the walls of macropores": You are absolutely right about that, the squeezing of liquid does not allow you to characterize correctly the material but your whole analysis is based on that. Why didn't you try to really dry these materials one way or another (freeze-drying, supercritical drying that has been proved to be the image of the wet gel or cryostructure), and then characterize it by the usual characterization methods, i.e. microscopy (TEM rather than optical, surface area measurements, X-ray tomography)?

- p. 6, l. 9-10: "the dependence of the degree of swelling of Ca-ACRS samples on the freezing temperature of the Na-ALG solution, in particular ": What is not really clear here is if you need for your planned applications a high degree of swelling or not? What are you looking for in your materials?

- p. 7, paragraph 2.2: micrographs presented in figure 5 have a very large scale (which is normal for optical microscopy). As already mentioned earlier, you should use an efficient drying method to perform TEM microscopy, and thus have a real picture of macropores in your materials, which is absolutely not the case here. As you mention, the direction of ice crystals is perpendicular to the cold surface on which the petri dish is set and you obviously obtain different pictures from top and bottom disks but you cannot conclude anything about the "size" of macropores in your pictures. Besides, you should also take pictures of the "side" of the disks to better observe the competing phenomena at work during the freezing process.

- p. 8, l.46-50 & Fig. 6: "photographs of Figure 6, where one can see the appearance of the test samples during their incubation in the above-indicated milieus. As a result, it was found that these spongy cryostructurates are stable not only in water (Fig. 6a-6d) and acidic medium (Fig. 6i-6l), where neither Ca-alginate nor alginic acid dissolve, but also in an excess of Na+-ions in physiological solution": The pictures are too small and the quality is not good enough to allow us to conclude something about the stability of the cryostructures. Besides, you only make qualitative observations: even of the macroscopic structure does not change, what happens at the microscopic level?

Experimental Section

- p. 10, l. 23-25: What is the overall mannuronic/guluronic ratio (and not only the content in blocks), as it has been shown to modify the physico-chemistry of alginate gels? Have you also determined the viscosity of your starting material, even if you did not determine it yourself, it is often given by alginate providers in the form of viscosity in a 1% solution? As you discussed it yourself, it is an extremely important parameter that can of course influence your end-material? Did you also think of using other types of alginate to check the influence of blocks, mannuronic/guluronic ratio, viscosity,..?

- p. 11, l. 7: Why did you choose to freeze the alginate solutions for only 1h, when a longer freezing time might have been beneficial to decrease the amount of unfrozen liquid microphase, even if it also decreases the size of pores? It could have been at least worth it to study the effect of freezing time.

- p. 11, l. 37: How long did the samples stayed at low temperature before taking NMR measurements? Is it the same time as for the materials synthesis?

- p. 11, l. 42: Did you estimate the amount of vancomycin loading on your materials at the end of your loading process?

Author Response

Here are several remarks to be considered:

Results and discussion

- p. 3, l. 10-14: "On the other hand, when the concentration of Na-ALG was above the specified range, a considerable increase in the viscosity of the stock polymer solution took place thus significantly complicating its dosing": What is this specified range? Besides, in numerous publications, when synthesizing alginate gels, foams or macroporous structures,... the starting aqueous alginate solutions concentrations are more often ranging between 1-5wt%, whereas you chose to use 0.1-0.5wt%, which is very low, and probably explains the very weak cryostructurates obtained at 0.1%. What is significantly more "complicated" to dose when using higher concentrations?

This reviewer’s remark contains the arithmetic mistake. We wrote (- p. 3, l. 10-14 in the initial manuscript): Preliminary experiments showed that calcium-alginate cryostructurates (Ca-ACRS) and alginic acid cryostructurates (H-ACRS) that possessed the properties convenient for further research can be formed from the sodium alginate (Na-ALG) aqueous solutions in the range of polymer concentrations from 10 to 50 g/L”. In the terms of weight concentrations this “specified” (has been changed for “indicated” in the revised manuscript) range for the w/v-concentration diapason of 10-50 g/L corresponds to ~1-5% values (more exactly, 0.99-4.76%) rather than 0.1-0.5wt% (i.e. one order lower) incorrectly indicated by the reviewer.

- p.3, l.27-28: " In turn, in ethanol, thus “tanned” Ca-ACRS and H-ACRS cryostructurates were tough, ": What does a "tanned" Ca-ARS mean exactly?

“Tanned” is the technological term usually used regarding the materials subjected to some chemical treatment in order to increase the stability. In our case, “tanned” meant “insolubilized” – we changed the word in the revised manuscript.

- p.3, l. 29: " As a result, a free fluid from the interconnected wide pores could be pressed out, for example, on a glass filter under vacuum (see ‘Experimental’)." : I do not understand, after reading the experimental section, how you can write that you have dried materials that still contain liquid..."This property of the water-swollen Ca-alginate and alginic acid cryostructurates, i.e. squeezing of capillary liquid by mechanical action, does not allow the correct measurement of the physico-mechanical characteristics of the swollen matrices, but it makes possible to evaluate the degree of swelling (Sw/w, g H2O/g polymer) of their polymeric phase, i.e. the walls of macropores": You are absolutely right about that, the squeezing of liquid does not allow you to characterize correctly the material but your whole analysis is based on that. Why didn't you try to really dry these materials one way or another (freeze-drying, supercritical drying that has been proved to be the image of the wet gel or cryostructure), and then characterize it by the usual characterization methods, i.e. microscopy (TEM rather than optical, surface area measurements, X-ray tomography)?

This reviewer’s remark contains two questions (blue text).

(i) First, we wrote: “(see ‘Experimental’)”. Let us see: “The swelling degree of the polymeric phase (the walls of macropores) in the wide-pore Ca-alginate and alginic acid cryostructurates was determined by the gravimetric method. To do this, free liquid was removed from the water-swollen sponges on a glass filter under vacuum of a water-jet pump. Thus obtained “pressed-out” wet samples were weighed and then dried at 110°C to a constant weight in an air thermostat”. This text means that after the forced removal of a free capillary liquid from the swollen sponges the “residual” wet matter was weighed and than finally dried at elevated temperature. This is a standard procedure, which is a rather experimentally simple.

(ii) The answer for the second question is as follows: The following phrase has been added to the revised manuscript at the end of ‘Introduction’ section: “… we mainly interested in the properties of wet and swollen alginate sponges since upon their use as the wound dressings, the materials operate in aqueous media”. Therefore, the studies of the properties of dried sponges did not include in the tasks of this particular work. The experiments suggested by the reviewer are an half, at least, of the PhD dissertation. At the same time, the present study is not the finish of our activity in this field, so, when it will be required in the course of continuing research, a more detailed characterization of the dried sponges will be performed. At present time, we are mainly focusing on the in vivo experiments in order to confirm the biomedical potential of these alginate-based materials. On the other hand, if such biotesting will result in the negative conclusions, the necessity of the dry matter studies is of doubt.

- p. 6, l. 9-10: "the dependence of the degree of swelling of Ca-ACRS samples on the freezing temperature of the Na-ALG solution, in particular ": What is not really clear here is if you need for your planned applications a high degree of swelling or not? What are you looking for in your materials?

We wanted to know how the freezing temperature can influence on the density of 3D network inside the pore walls of wide porous alginate and alginic cryostructurates, since such effect is known for other cryogenically-structured matrices based on other polymer precursors (e.g., see refs. 36-48 in the revised manuscript), and the swelling extent of the pore walls is an indicator of such density (the cross-linking degree). In addition, regarding the application of the drug-loaded alginate sponges: the ratio of the loaded drug amount inside the capillary pores and within the polymeric phase of pore walls will depend on their (the walls) swelling extent. This ratio should influence on the drug release kinetics – quick from the intra-pore space and somewhat slower from the pore walls. Such release experiments are in progress now in our Lab; the experiments are connected directly with the in vivo testing, which are carried out by the biologists.

- p. 7, paragraph 2.2: micrographs presented in figure 5 have a very large scale (which is normal for optical microscopy). As already mentioned earlier, you should use an efficient drying method to perform TEM microscopy, and thus have a real picture of macropores in your materials, which is absolutely not the case here. As you mention, the direction of ice crystals is perpendicular to the cold surface on which the petri dish is set and you obviously obtain different pictures from top and bottom disks but you cannot conclude anything about the "size" of macropores in your pictures. Besides, you should also take pictures of the "side" of the disks to better observe the competing phenomena at work during the freezing process.

Surprisingly that the majority of the reviewer’s remarks are related to the dry matters (obviously, solid materials are the subject of his/her professional interests), whereas this particular research has been dealt with the swollen cryostructurates, as well as with their behavior in the liquid media. Such direction of our studied has been tightly connected with the conditions of possible medical implemental of the developed alginate sponges.

It is clear that common TEM and SEM techniques can not be used for similar swollen polymeric matrices, while the cryo-electronic microscopy gives the structural information only on the surface of the respective samples in the frozen state.

We disagree with reviewer’s opinion that using of optical microscopy does not allow evaluating the size of macropores at the top and bottom sides of “our” alginate disks. Optical micrographs of swollen sponges clearly show the character of shape and dimensions of the macropores of these areas of such soft water-swollen material. Of course, it will be interesting to obtain the image “from the side of the disks”, but it could be done only using thin sections in the direction orthogonal to the top and bottom surfaces. Unfortunately, it turned out that neither cryomicrotome, nor inclusion in paraffin with subsequent sectioning with usual microtome, did not give satisfactory results for the swollen sponges – the samples were deformed (rumpled) under the action of microtome knife. Therefore, since we still have no the quantitative data on the size of pores inside the disks, only the size of pores on the top and bottom surfaces of the water-swollen sponges are given and discussed in this paper.

- p. 8, l.46-50 & Fig. 6: "photographs of Figure 6, where one can see the appearance of the test samples during their incubation in the above-indicated milieus. As a result, it was found that these spongy cryostructurates are stable not only in water (Fig. 6a-6d) and acidic medium (Fig. 6i-6l), where neither Ca-alginate nor alginic acid dissolve, but also in an excess of Na+-ions in physiological solution": The pictures are too small and the quality is not good enough to allow us to conclude something about the stability of the cryostructures. Besides, you only make qualitative observations: even of the macroscopic structure does not change, what happens at the microscopic level?

In the initial version of our manuscript submitted to ‘GELS’ Figure 6 was located in the whole horizontal page, where the photographs were considerably larger and clearer. We want to ask the production editor to come back, if possible, to the location of this figure spread within the whole page.

            As for the question about the changes with the disks at the microscopic level during incubation the samples in various liquids, this problem requires special separate investigation, which was not the target of this particular study. The goal of the pictures in Fig. 6 is to demonstrate the macroscopic behavior of the elaborated alginate sponges in such media. Again, this has been done in a view of our applied interests.

Experimental Section

- p. 10, l. 23-25: What is the overall mannuronic/guluronic ratio (and not only the content in blocks), as it has been shown to modify the physico-chemistry of alginate gels? Have you also determined the viscosity of your starting material, even if you did not determine it yourself, it is often given by alginate providers in the form of viscosity in a 1% solution? As you discussed it yourself, it is an extremely important parameter that can of course influence your end-material? Did you also think of using other types of alginate to check the influence of blocks, mannuronic/guluronic ratio, viscosity,..?

We added to ‘Experimental’ the data on the viscosity of 1% aqueous solution of Na-alginate, which was used for the preparation of cryogenically-structured sponges. As for the use of other alginate types, it is planned (when the primary in vivo tests will be completed) to focus on the alginate of ‘medical brands’. G and M content, as well as the overall characterization, will be performed in such a case, and, probably, for the polymers from various manufacturers. Certainly, we understand well the necessity of such characterization. 

- p. 11, l. 7: Why did you choose to freeze the alginate solutions for only 1h, when a longer freezing time might have been beneficial to decrease the amount of unfrozen liquid microphase, even if it also decreases the size of pores? It could have been at least worth it to study the effect of freezing time.

We found that freezing duration of 1h-long is enough to guarantee the freezing completion for the samples of Na-alginate aqueous solutions poured in the small Petri dishes. Of course, when the sponges of higher size will be considered, we will check the influence of the freezing duration on the properties of the resultant cryostructurates.

- p. 11, l. 37: How long did the samples stayed at low temperature before taking NMR measurements? Is it the same time as for the materials synthesis?

Yes, the time was virtually the same with the variation no more than about 5 min.

- p. 11, l. 42: Did you estimate the amount of vancomycin loading on your materials at the end of your loading process?

Many thanks for this remark – we simply forget to include such data in the primary manuscript. Now, this mistake has been corrected (see ‘Experimental’ in the revised manuscript).

Round 2

Reviewer 2 Report

The authors have answered satisfactorily to my concerns, and therefore the paper can be accepted to be published in Gels, providing the authors make very small modifications:

- I agree with you that your interest here lies mainly in the wet state of the sponge, and not the dry material. I was simply pointing out that a full characterization of the materials, which would be nice from a scientific point of view, would require analyses only accessible when materials are dry. If the biomedical potential of these sponges is confirmed you probably will have to think about it.

- p.6, l.9-10: Concerning my remark on the interest of swelling or not for your planned applications, the precisions you give are really interesting (effect of swelling on the ratio of loaded drug), and I think they should be included in the main text of the publication for everybody to read, and not only for me.

- I agree with you that figure 6 should be located back on a full horizontal page for a better reading.

Author Response

We included the data suggested by the reviewer in the revised manuscript; these text inclusions are indicated by the blue color.